# Spring Viremia of Carp Virus Infection Induces Carp IL-10 Expression, Both In Vitro and In Vivo

**DOI:** 10.3390/microorganisms11112812

**Published:** 2023-11-20

**Authors:** Ping Ouyang, Yu Tao, Wenyan Wei, Qiunan Li, Shuya Liu, Yongqiang Ren, Xiaoli Huang, Defang Chen, Yi Geng

**Affiliations:** 1Department of Basic Veterinary, College of Veterinary Medicine, Sichuan Agricultural University, Chengdu 611130, China; taoyu4030@163.com (Y.T.); liqiunan1216@163.com (Q.L.); liushuya0927@163.com (S.L.); ryq3280@163.com (Y.R.); gengyisicau@126.com (Y.G.); 2Chengdu Academy of Agriculture and Forestry Sciences, Chengdu 611130, China; feitianyan2008@163.com; 3Department of Aquaculture, College of Animal Science and Technology, Sichuan Agricultural University, Chengdu 611130, China; hxlscau@126.com (X.H.); chendf_sicau@126.com (D.C.)

**Keywords:** spring viremia of carp virus (SVCV), interleukin 10 (IL-10), JAK-STAT, NF-κB, p38MAPK

## Abstract

Interleukin-10 (IL-10) is a pleiotropic cytokine with both immune enhancement and immunosuppression activities, but the main role is immunosuppression and anti-inflammatory ability. In order to use the immunosuppressive function of IL-10, many viruses, such as SARS-CoV-2, hepatitis B virus and EB virus, can evade the host’s immune surveillance and clearance by increasing the expression of host IL-10. However, it has not been reported whether the aquatic animal infection virus can upregulate the expression of host IL-10 and the mechanisms are still unknown. Spring viremia of carp (SVC) is a fatal viral disease for many fish species and is caused by spring viremia of carp virus (SVCV). This disease has caused significant economic losses in the aquaculture industry worldwide. In this study, the expression of carp IL-10 with or without infection of SVCV in epithelioma papulosum cyprinid (EPC) cells, carp head kidney (cHK) primary cells and common carp tissues were analyzed using RT-PCR and ELISA. The results show that SVCV infection induced carp IL-10 mRNA and protein expression, both in vitro and in vivo. However, the upregulation of carp IL-10 by SVCV was hindered by specific inhibitors of the JAK inhibitor (CP-690550), STAT3 inhibitor (STA-21), NF-κB inhibitor (BAY11-7082) and p38 MAPK (mitogen-activated protein kinase) inhibitor (SB202190), but not JNK inhibitor (SP600125). Furthermore, the results demonstrated that JAK1, JAK2, JAK3, TYK2 and STAT5 played important roles in carp IL-10 production induced by SVCV infection. Taken together, SVCV infection significantly induced carp IL-10 expression and the upregulation trigged in JAK-STAT, NF-κB and p38MAPK pathways. To our knowledge, this is the first time that a fish infection virus upregulated the host IL-10 expression through the JAK-STAT, NF-κB and p38MAPK pathways. Altogether, fish viruses may have a similar mechanism as human or other mammalian viruses to escape host immune surveillance and clearance.

## 1. Introduction

Spring viremia of carp virus (SVCV), which is the causative agent of spring viremia of carp (SVC), was classified as a member of the family *Rhabdoviridae* and belongs to the genus *Vesiculovirus* [1]. The genome of SVCV is a linear single-stranded negative RNA. SVC has been registered in the list of contagious diseases notifiable to the World Organization for Animal Health (WOAH) because of its significant risk and harmfulness. It was also recognized as one of the class II diseases by the Animal Epidemic Prevention Law of the People’s Republic of China in 2022. All farmed fish in SVC outbreak areas have to be euthanized to control this disease [2,3]. 

Fiorentino et al. found that activated mouse Th2 cells produce an active substance that can inhibit the activation of Th1 cells and the production of cytokines in 1989 [4]. It was named cytokine synthesis inhibitor (CSIF), and then it was uniformly renamed interleukin 10 (IL-10). IL-10 is produced by almost all leukocyte subtypes, though CD4+ T cells and monocytes/macrophages are considered the most important sources of this cytokine [5]. IL-10 is a multifunctional cytokine with both immune enhancement and immunosuppression activities, but its main function is immunosuppression. By inhibiting the function of helper cells (such as macrophages and monocytes), IL-10 indirectly inhibits a variety of target cells of the immune system, such as T cells and B cells, transmits immune information and participates in the immune response of the body. Monocytes/macrophages are recognized as IL-10’s main target to directly inhibit the synthesis of pro-inflammatory cytokines, reactive radical species, and surface expression of molecules involved in antigen presentation and phagocytosis [6]. Several viruses were shown to upregulate the expression of IL-10 via the suppression of immune functions, such as HCV [7], HIV [8] and SARS-CoV-2 [9]. To date, viral IL-10s have been reported with at least 21 viruses, including two fish infection viruses (Cyprinid herpesvirus 3 and Anguillid herpesvirus 1) [10]. Infectious pancreatic necrosis virus (IPNV) infection induces an increase in the expression of IL-10 mRNA levels in the spleen and head kidney in Atlantic salmon [11,12]. The expression of IL-10 was increased in Atlantic salmon peripheral blood at the late stage of the infection with infectious salmon anemia virus (ISAV) [13]. Bjørgen et al. 2020 showed a significant downregulation of IL-10 in the red muscle changes, while an increase in IL-10 was present in the melanized muscle changes associated with Piscine orthoreovirus (PRV) infection [12]. However, the mechanism of IL-10 upregulation induced by the aquatic animal infection virus has not been studied.

Fish IL-10s have been cloned, including those in Cyprinus carpio, Carassius auratus, Ctenopharyngodon idella, Danio rerio, Oncorhynchus mykiss, Dicentrarchus labrax, Labeo rohita, Takifugu rubripes and other bony fish [14,15,16,17,18,19,20]. The amino acid sequence of Cyprinus carpio IL-10 has high homology with that of Carassius auratus, Ctenopharyngodon idella and Danio rerio [18]. The phagocytosis of fish IL-10 and pmt-38 cells was similar to that of mammalian IL-10 and pmt-38 cells, which can also promote the production of respiratory and memory cells. After carp IL-10 binds to its receptor, the signal inhibits the production of inflammatory cytokines through the JAK/STAT3 pathway [6]. However, the underlying regulatory mechanisms involved in IL-10 induction during SVCV infection have not been explored.

In the present study, the kinetic expression of IL-10 with or without SVCV infection was investigated in cells and common carp tissues. The results demonstrate that SVCV infection significantly upregulated IL-10 mRNA and protein expression both in vitro and in vivo. SVCV infection upregulated the expression of carp IL-10 triggered JAK-STAT, NF-κB and p38MAPK pathway, but not the JNK pathway. Altogether, fish viruses may have a similar mechanism of immune evasion to that of humans or other mammalian viruses. These results offer an insight into the complexity of the virus–cell interactions in aquatic animals. 

## 2. Materials and Methods

### 2.1. Viruses, Cells and Fish

The SVCV strain was isolated from the spleen and kidney mixture of a diseased *Percocypris pingi* by using Epithelioma papulosum cyprini (EPC) cells in Chengdu, China, in 2018 [21]. EPC cells were cultured at 25 °C in Medium 199 (HyClone, South Logan, UT, USA) maintained with 10% fetal bovine serum (Gibco, Grand Island, NY, USA) and 100 U/mL Penicillin–Streptomycin Liquid (Solarbio, Beijing, China). Four common carp (body weight of approximately 750 ± 50 g) were obtained from a local farm in Chengdu, Sichuan Province, and used for carp head kidney (cHK) primary cell isolation. The primary cell culture was performed as previously described [22]. The head kidney was minced into small pieces with a sterile blade. The mixture was pressed through 100 mesh sieves to remove incompletely digested tissues. The mixture was centrifuged at 1500 rpm for 30 min at 4 °C. The pellet was re-suspended in a DMEM medium containing 0.1% fetal bovine serum (FBS) and two antibiotics. Finally, the suspension was transferred into 25 cm tissue culture flasks and maintained at 25 °C. The morphology of the attached cells was observed under a phase contrast microscope. And then the isolated cHKs were subjected to further experiments.

Common carp (body weight of approximately 100 ± 10 g) were obtained from a local farm in Chengdu City, Sichuan Province. Common carp were kept in 60 L tanks for 7 days for adaption. During this period, the dissolved oxygen was 6.5 ± 0.5 mg/L, pH was 6.5–7.0, ammonia nitrogen was 0.054 ± 0.028 mg/L and water temperature was 16 ± 1 °C. A total of 60 healthy common carp were randomly divided into two groups, including a control group and an SVCV challenge group. Microbiological, parasitical and clinical examinations of the carp demonstrated that these fish were fully healthy. All experiments were performed under anesthesia and all efforts were made to minimize the suffering of fish. The experimental protocols were performed in accordance with the Animal Experiment Guidelines of Sichuan Agricultural University under permit number TY-S2019203012. 

### 2.2. Reagents 

JAK inhibitor (CP-690550), STAT3 inhibitor (STA-21), NF-κB inhibitor (BAY11-7082) and MAPK inhibitors (SP600125 and SB202190) were purchased from TargetMol (Shanghai, China). These inhibitors were dissolved in DMSO (Amresco, Solon, OH, USA) prior to use. The effects of inhibitors on the toxicity of cells were determined using a CCK-8 assay (BBI, Shanghai, China).

### 2.3. Stimulation of EPC and cHK Cells with SVCV Infection 

Both EPC and cHK cells were infected with SVCV at a multiplicity of infection (MOI) of 0.5. The cells were incubated with the virus for 2 h at 25 °C. Then, the medium was changed to a fresh growth medium. The cells were incubated and the samples were collected at the indicated time points (3 h, 6 h, 9 h, 12 h, 24 h and 48 h). Total RNA was isolated from cells by using an RNA Nano 6000 Assay Kit of the Bioanalyzer 2100 system (Agilent Technologies, Santa Clara, CA, USA). The quality and quantity of the mRNA were determined using a NanoDrop (Thermo Fisher Scientific, Waltham, MA, USA) at 260 and 280 nm wavelengths. The cDNA was synthesized from 1 μg of RNA by using Hiscript III RT SuperMix for qPCR (+gDNA wiper) (Vazyme, Nanjing, China). Protein extraction was conducted using a total protein extraction kit (BC3640, Solarbio, Beijing, China) according to the manufacturer’s manual. The concentration of protein was measured using the BCA protein assay kit (Solarbio, Beijing, China). 

### 2.4. SVCV Inoculation of Carp

Common carp (body weight of approximately 100 ± 10 g) were fed 7 days of adaptive feeding using recycled water before the experiment. The water temperature was maintained at 16 ± 1 °C and the dissolved oxygen was 6.5 ± 0.5 mg/L. Five fish were randomly selected for nested RT-PCR to ensure that the carp were free of SVCV infection. Sixty fish were randomly divided into two groups. The experimental group of fish was infected using SVCV with TCID_50_/mL by using intraperitoneal injection. The other group of fish was injected with the same volume of cell culture medium used as a control. Gills, head kidney, kidney, spleen and intestine were collected at 3 h, 24 h, 72 h and 144 h post-infection for mRNA and protein isolation.

### 2.5. Inhibition of Signal Transduction Pathways

EPC cells were pre-incubated with DMSO, CP-690550 (100 nM), STA-21 (30 μM), BAY11-7082 (10 μM), SB202190 (100 nM) or SP600125 (100 nM) for 3 h at 25 °C. Then, the cells were infected with SVCV at an MOI of 0.5 for 1 h, and then inhibitors were added. Cells were harvested for mRNA and protein isolation at 6 h, 12 h, 24 h and 48 h post-infection. 

### 2.6. Quantification of Carp IL-10 Gene Expression Using RT-qPCR

Cytoplasmic RNA from cells and tissues was analyzed using the RNeasy Mini kit with on-column DNase I digestion. cDNA was synthetized from 1 μg of RNA using HiScriptR III RT SuperMix (Vazyme, Nanjing, China) according to the manufacturer’s instructions. And reverse transcription polymerase chain reaction was performed using ChamQ Universal SYBR qPCR Master Mix (Vazyme, Nanjing, China). PCRs were performed with the primers listed in Table 1. Each cDNA sample was performed in triplicate. The RT-PCR master mix was prepared as follows: SYBR Green Supermix, 200 nM of each primer, 5 μL diluted cDNA and sterile water to a final volume of 20 μL. The amplification program included an initial denaturation at 95 °C for 30 s, followed by 40 cycles with denaturation at 95 °C for 5 s, annealing at 60 °C for 30 s and elongation at 72 °C for 30 s. Ct values from the carp gene expression were normalized to the Ct levels of β-actin. The expressions of analyzed genes were calculated by using the 2^−ΔΔCT^ method. The assays were performed on a real-time fluorescence quantitative PCR System (Bio-Rad, Hercules, CA, USA).

### 2.7. Quantification of IL-10 Protein Expression Using ELISA 

Protein expressions of carp IL-10 were determined using a commercially available ELISA kit for carp IL-10 (R&D systems, Shanghai Keshun Science and Technology (Shanghai, China), product no. YX-091210C). The experimental procedure was performed according to the manual’s instructions. The 50 µL diluted standard substance and samples were added to a 96-well plate and incubated at 37 °C for 60 min. After washing 5 times, 50 µL reagent A and 50 µL reagent B were added to each well, followed by incubation at 37 °C for 10 min. Finally, a 50 µL stop solution was added to each well and the absorbance was measured at 450 nm using a spectrophotometer (Thermo Fisher Scientific, Waltham, MA, USA). A standard curve was prepared to determine the sample concentration based on the optical density value.

### 2.8. Statistical Analysis

All experiments were performed at least three times. Data were analyzed using GraphPad Prism 6 (GraphPad Software, San Diego, CA, USA). One-way ANOVA was used for multiple comparisons, followed by Tukey’s test. In all cases, asterisks (*) indicate significant differences between the two comparison groups. Statistical significance was considered as * *p* < 0.05, ** *p* < 0.01, *** *p* < 0.001 and **** *p* < 0.0001.

## 3. Results

### 3.1. SVCV Infection Induced Carp IL-10 Production at mRNA and Protein Levels in EPC and cHK Cells

RT-PCR and ELISA were performed to detect IL-10 mRNA and protein expression in cHK and EPC cells at different time points, respectively. Carp IL-10 mRNA was significantly upregulated in cHK cells at 3 hpi and 6 hpi with SVCV. There were no significant differences between the mock and infected cHK cells at 9 hpi, 12 hpi, 24 hpi and 48 hpi (Figure 1A). The carp IL-10 mRNA was upregulated in EPC cells at 12 hpi with SVCV. The expression of carp IL-10 mRNA reached the peak at 24 hpi and declined at 48 hpi (Figure 1B). In addition, the SVCV infection induced an earlier expression of IL-10 in the cHK compared with the EPC cells. The protein expression manner of IL-10 was consistent with mRNA. The carp IL-10 protein expression was significantly upregulated in the cHK cells at 3 hpi and 6 hpi with SVCV (Figure 1C). And the protein expression of IL-10 was significantly upregulated at 12 hpi in the EPC cells (Figure 1D). Thus, these results suggest that the SVCV can stimulate carp IL-10 production in cHK cells and EPC in vitro.

### 3.2. SVCV Infection Induced Carp IL-10 Production at mRNA and Protein Levels in Carp Tissues

To investigate the kinetic expression of IL-10 in SVCV-infected carp, the fish were infected with or without SVCV. The expression of carp IL-10 mRNA was significantly increased after the SVCV infection in the head kidney, gills and spleen. Compared with the control group, there was no significant difference in the kidney and intestine. However, the carp IL-10 mRNA expression was increased in the kidney and intestine as well (Figure 2). The carp IL-10 mRNA in the gill tissue was rapidly upregulated at 3 hpi and maintained at a stable expression level at 24 hpi, 72 hpi and 144 hpi. The expression of the carp IL-10 gene in gills was the highest among the other five tissues (Figure 2A). The carp IL-10 mRNA in splenic tissue increased and then decreased. The carp IL-10 mRNA expression level peak appeared at 24 hpi and lasted until 72 hpi. There was no significant difference in the expression level of the IL-10 gene in spleen tissues at 3 hpi and 144 hpi (Figure 2B). The carp IL-10 mRNA was rapidly upregulated in the head kidney at 24 hpi, 72 hpi and 144 hpi (Figure 2C). Compared with the control group, there was no significant difference in the expression of IL-10 mRNA in the kidney and intestine at the indicated time points (Figure 2D,E). 

The protein expression manner of carp IL-10 was consistent with the mRNA expression (Figure 3). The carp IL-10 protein expression rapidly increased in gill tissue at 3 hpi, and maintained a high expression level at 24 hpi, 72 hpi and 144 hpi (Figure 3A). Compared with the control group, the expression of carp IL-10 protein in the spleen was significantly different at 24 hpi (Figure 3B) and showed an increasing trend at other time points (Figure 3B). In the head kidney, the carp IL-10 protein expression was upregulated at 24 hpi, 72 hpi and 144 hpi in the infection group (Figure 3C). There was no significant difference in the IL-10 protein expression in the kidney and intestinal tissues, which was consistent with the gene expression (Figure 3D,E).

### 3.3. Inhibition of JAK-STAT, NF-κB and p38MAPK Pathways Hindered the Upregulation of Carp IL-10

To investigate the signaling pathways involved in the stimulation of carp IL-10 via SVCV infection, five specific inhibitors were used in this study, namely, JAK inhibitor (CP-690550), STAT3 inhibitor (STA-21), NF-κB inhibitor (BAY11-7082), p38 inhibitor (SB202190) and JNK inhibitor (SP600125). The cytotoxicity of all the inhibitors was determined using a CCK-8 assay. All concentrations of the inhibitors used in this study neither caused detectable cell death nor significantly altered SVCV replication in EPC cells EPC cells were infected with SVCV in combination with or without inhibitors at 6 h, 12 h, 24 h and 48 h. The total RNA and protein were isolated and detected using RT-qPCR and ELISA, respectively. The results show that CP-690550, STA-21, BAY11-7082 and SB202190 downregulated the carp IL-10 production, which was upregulated by SVCV infection (Figure 4). No significant upregulation of the carp IL-10 gene expression was observed in the SP600125 inhibitor treatment group. These data suggest that the upregulation of carp IL-10 expression was involved in the JAK-STAT signaling pathway, NF-κB pathway and p38 pathway. However, the IL-10 upregulation was not triggered by the JNK signaling pathway (Figure 4).

### 3.4. JAK1, JAK2, JAK3, TYK2, STAT5 and STAT6 Were Involved in SVCV-Induced Carp IL-10 Expression

We were interested in determining which pathway factors were involved in SVCV-induced IL-10 production. The gene expressions of JAK1, JAK2, JAK3, TYK2, STAT5 and STAT6 were analyzed using RT-qPCR. The gene expressions of JAK1, JAK2, JAK3, TYK2 and STAT5 were inhibited by using the JAK inhibitor. However, there was no significant difference in the expression of STAT6 (Figure 5). The gene expressions of JAK2, JAK3 and TYK2 were inhibited by using the NF-κB signal pathway inhibitor. The P38 signal pathways inhibitor only inhibited the gene expressions of JAK3 and TYK2. The results showed that the gene expression of JAK3 was inhibited by the treatment of the STAT3 inhibitor, but there was no effect on other factors. However, the JNK inhibitor had no effect on the expressions of the studied factors (Figure 5). Based on the above results, SVCV-infected EPC cells caused the upregulation of carp IL-10, which may have triggered the JAK-TYK2-STAT5 and JAK3-STAT3 pathways. Moreover, NF-κB and p38 signaling pathways may have also involved the carp IL-10 upregulation via SVCV infection (Figure 6).

## 4. Discussions

In this study, the expression manner of carp IL-10 on cells and tissues using SVCV infection was investigated. Previous studies showed that a variety of cells, such as macrophages, monocytes, leukocytes and various T cell subsets, can produce IL-10, but macrophages are the main source, and some tissue cells cannot produce IL-10 [20]. In EPC cells and cHK cells, IL-10 was significantly upregulated after SVCV infection at 12 h and 3 h post-infection, respectively. This result was consistent with the previous studies [20]. The expression of IL-10 mRNA was upregulated in LPS treated in head kidney leukocytes and *Aeromonas hydrophila*-infected fish, and the upregulation time of IL-10 expression was later than that of the pro-inflammatory factor TNF-α [25,26]. It was speculated that IL-10 may be involved in the regulation of pro-inflammatory factor expression [18,27].

Carp IL-10 expression was upregulated by the infection of SVCV in the gills, spleen and head kidney, but not in the kidney and intestine at the indicated time points. The IL-10 expression was upregulated when stimulated by LPS in the kidney, gill and intestine, as demonstrated in adult zebrafish by using RT-PCR [27]. Previous studies found that IL-10 was significantly upregulated in peripheral blood after viral infection [13,28]. Huo et al. showed the upregulated expression of IL-10 in the spleen after infection with infectious spleen and kidney necrosis virus (ISKNV) [29]. Wang et al. found that IL-10 expression in six tissues of *Oplegnathus punctatus* was upregulated during spotted knifejaw iridovirus (SKIV) infection, and the expression of IL-10 in the liver was elevated throughout the infection period, indicating that IL-10 plays an important role in pathogenic infection [30].

Fish virus infections that can upregulate the expression of IL-10 have been reported before. The upregulation of IL-10 expression with SVCV infection in cHK cells was presented as early as 3 h post-infection. While the upregulation of IL-10 expression was shown at 12 hpi in the EPC cells, the expression manner was later than in the cHK cells. This may have been because IL-10 is mainly secreted by immune cells. In the SVCV-challenged common carp, the IL-10 expression was significantly increased at 3 hpi in the gills and the peak appeared at 24 hpi. Meanwhile, the IL-10 expression significantly increased at 24 hpi in the spleen and head kidney, respectively. Wang et al. reported that the *Oplegnathus punctatus* IL-10 expression was induced via infection with SKIV at 4 hpi in the gills and 1 dpi in the spleen. Our results were consistent with the previous studies [30]. Ingerslev et al. found a significant upregulation 24 days after intraperitoneal challenge with IPNV [13]. IL-10 is known as an inhibitor of interferon-γ in mice, and the upregulation of IL-10 could potentially be a mechanism to help the virus escape the host immune surveillance, but further research needs to be done in order to clarify the function(s) of IL-10 in fish viral infection.

The JAK-STAT pathway is one of the important signaling pathways downstream of cytokine receptors [31]. In mammals, IL-10 modulates the immune response by activating IL-10R, then triggering STAT3 phosphorylation, dimerization and translocation to the nucleus [32,33]. Many studies reported that IL-10 was primarily mediated by its interaction with IL-10 receptor, which activated the JAK1-TYK2-STAT3 pathway leading to STAT3-mediated transcription of genes that limit the inflammatory response [34]. Studies reported that IL-10 signals through crfb7 (homologous with IL-10R1) and crfb4 (homologous with IL-10R2) to activate the STAT3 signaling pathway in carp [35]. In this study, the results show that the SVCV-induced IL-10 expression in EPC cells triggered the JAK-STAT signaling pathway. 

Successful engagement of the IL-10 receptor complex subsequently activates distinct JAK-STAT pathways and downstream signaling events that converge through various mechanisms to influence nuclear transcriptional events, such as those mediated by NF-κB [36]. Previous studies showed that fish NF-κB plays a role in the immune response by regulating the expression of IL-1β [16], IL-12 [37] and other genes. Therefore, it is very important to explore whether the SVCV-infection-upregulated carp IL-10 triggers the NF-κB signal pathway. MAPK subfamilies play important roles in virus-induced IL-10 expression and different viruses utilize different MAPK pathways [25,38]. Agrawal S et al. reported that leptin activates human B cells to secrete IL-10 via the activation of the JAK2/STAT3 and p38MAPK/ERK1/2 signaling pathways [39]. Furthermore, treatment with specific inhibitors or siRNA knockdown assays demonstrated that p38 MAPK (mitogen-activated protein kinase) was required for PRRSV-induced IL-10 [12]. In this study, the results show that the p38 MAPK pathway was involved in SVCV-induced IL-10 expression, but SVCV infection in EPC cells might not trigger the JNK pathway.

JAK1, JAK2, JAK3, TYK2, STAT5 and STAT6 factors play important roles in the JAK-STAT signal pathway. The results of this study show that SVCV-induced IL-10 expression depended on the expressions of the cytokines JAK1, JAK2, JAK3, TYK2 and STAT5, but not STAT6. Therefore, the JAK/STAT-NF-κB–p38MAPK signaling cascade may be involved in SVCV-induced carp IL-10 expression. Moreover, the JAK-TYK2-STAT5 and JAK3-STAT3 pathways might be the two main signaling pathways in the upregulation of carp IL-10 induced by SVCV infection. To our knowledge, this is the first time the signal pathway involved in carp IL-10 regulation stimulated by SVCV infection has been analyzed. The results demonstrate that the regulation of carp IL-10 was involved in more than one signaling pathway, which enriches the study of fish interleukin-10 signaling pathways.

## Figures and Tables

**Figure 1 microorganisms-11-02812-f001:**
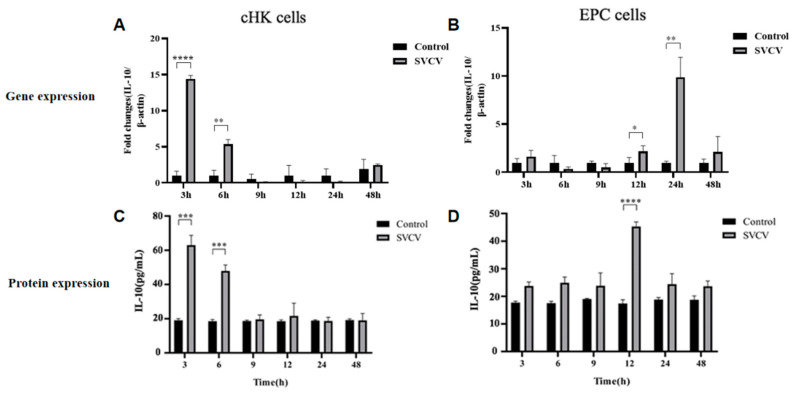
SVCV infection induced carp IL-10 expression in cHK and EPC cells. cHK and EPC cells were infected with SVCV at multiplicity of infection (MOI) of 0.5 or mock infected. The mock-infected cells were used as control. Cells were harvested at 3 h, 6 h, 9 h, 12 h, 24 h and 48 h post-infection. RT-PCR was performed to evaluate IL-10 mRNA expression level (**A**,**B**). ELISA was used to evaluate IL-10 protein expression (**C**,**D**). The results represent mean ± SD of three independent experiments. Statistical significance was considered as * *p* < 0.05, ** *p* < 0.01, *** *p* < 0.001 and **** *p* < 0.0001.

**Figure 2 microorganisms-11-02812-f002:**
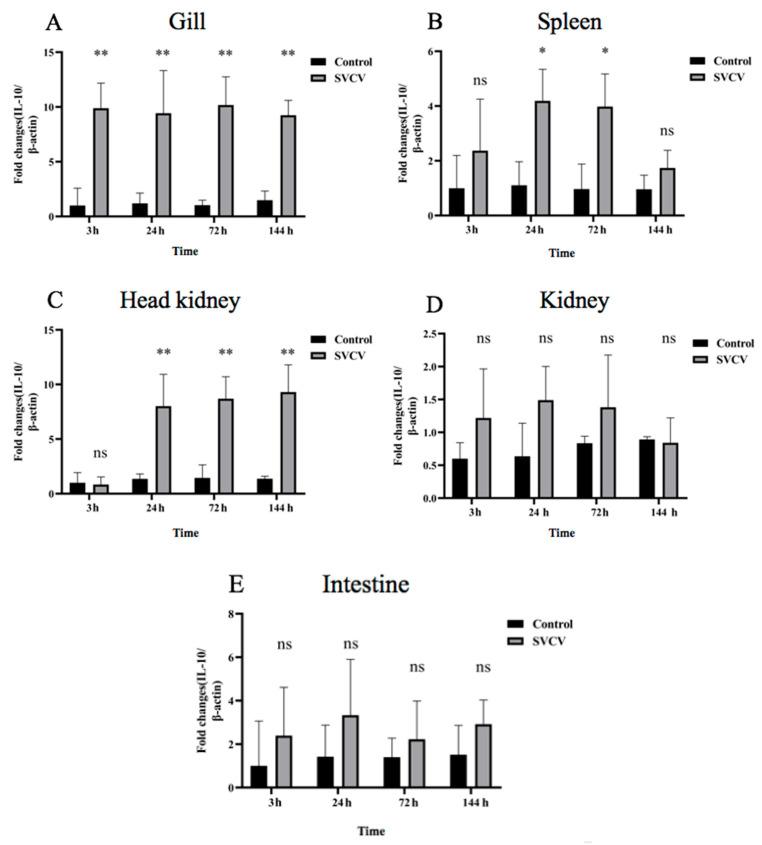
The expression of carp IL-10 mRNA was upregulated in SVCV-infected tissues at different time points. (**A**) The expression of carp IL-10 mRNA in gills; (**B**) the expression of carp IL-10 mRNA in spleen; (**C**) the expression of carp IL-10 mRNA in head kidney; (**D**) the expression of carp IL-10 mRNA in kidney; (**E**) the expression of carp IL-10 mRNA in intestine. Uninfected fish tissues were used as control. Data are presented as the mean ± SD of three replicates. Asterisk (*) indicates significant difference between SVCV-infected and uninfected groups, where * indicates *p* < 0.05 and ** indicates *p* < 0.01, “ns” indicates no significant difference *p* > 0.05.

**Figure 3 microorganisms-11-02812-f003:**
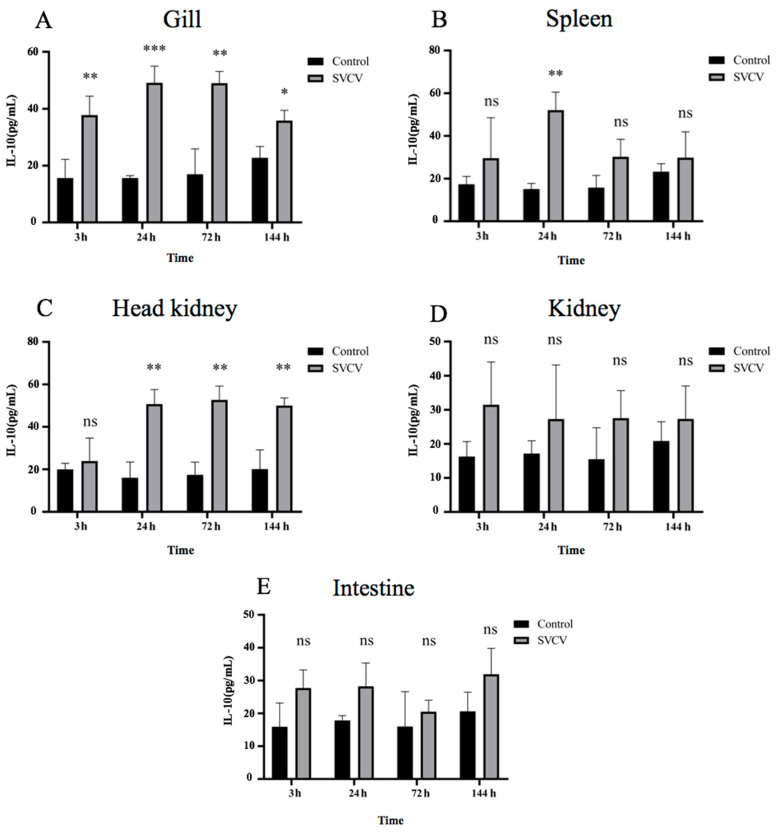
The protein expression of carp IL-10 was upregulated in SVCV-infected tissues with or without SVCV infection at different time points. (**A**) Carp IL-10 protein expression in gill tissue; (**B**) carp IL-10 protein expression in spleen tissue; (**C**) carp IL-10 protein expression in head kidney tissue; (**D**) carp IL-10 protein expression in kidney tissue; (**E**) carp IL-10 protein expressions in intestinal tissue. Uninfected fish tissues were used as control. Asterisk (*) indicates significant difference between SVCV-infected and uninfected groups, where * indicates *p* < 0.05; ** indicates *p* < 0.01 and *** indicates *p* < 0.001, ns indicates no significant difference *p* > 0.05.

**Figure 4 microorganisms-11-02812-f004:**
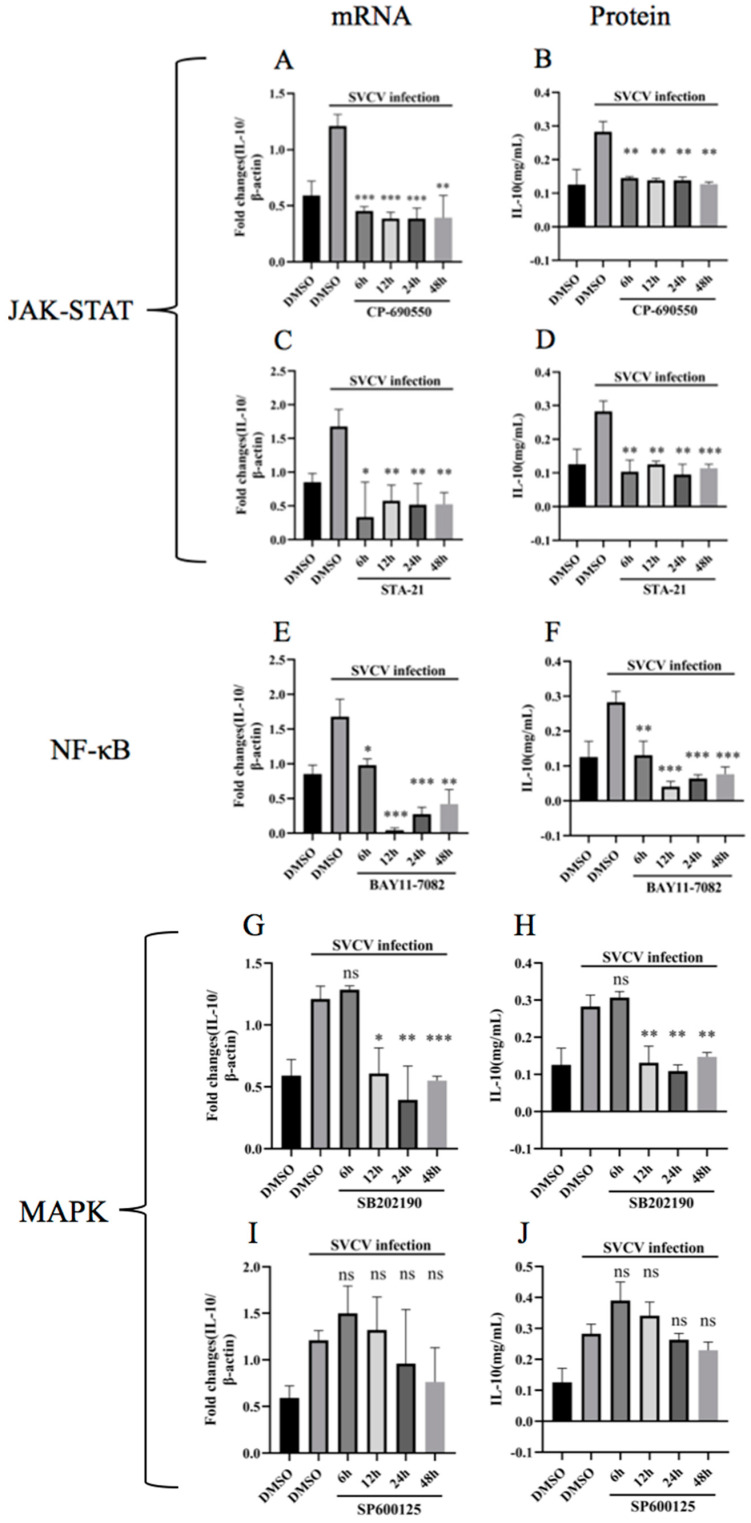
JAK-STAT, NF-κB and p38 signaling pathways played important roles in SVCV-induced carp IL-10 production. EPC cells were infected or uninfected with SVCV at MOI of 0.5, followed by treatment with JAK inhibitor CP-690550 (**A**,**B**), STAT3 inhibitor STA-21 (**C**,**D**), NF-κB inhibitor BAY11-7082 (**E**,**F**), p38 inhibitor SB202190 (**G**,**H**), JNK inhibitor SP600125 (**I**,**J**) or DMSO for 6 h, 12 h, 24 h and 48 h post-infection. The expression of carp IL-10 mRNA and protein were analyzed using RT-PCR and ELISA, respectively. Data are presented as the mean ± SD of three independent experiments. * *p* < 0.05, ** *p* < 0.01, *** *p* < 0.001 and, ns indicates no significant difference *p* > 0.05 compared with DMSO-treated plus SVCV-infected cells.

**Figure 5 microorganisms-11-02812-f005:**
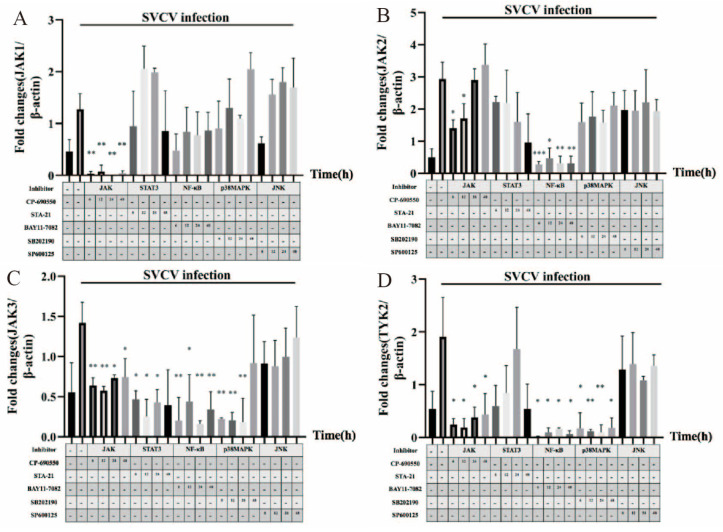
(**A**–**F**): Gene expressions of JAK1, JAK2, JAK3, TYK2, STAT5 and STAT6 were analyzed using RT-PCR in EPC cells. EPC cells were infected or uninfected with SVCV at MOI of 0.5, followed by treatment with CP-690550, STA-21, BAY11-7082, SB202190, SP600125 and DMSO, respectively, at 6 h, 12 h, 24 h and 48 h post-infection. The gene expressions were analyzed using RT-PCR. Data are presented as the mean ± SD of three independent experiments. * *p* < 0.05, ** *p* < 0.01, *** *p* < 0.001 and **** *p* < 0.0001 compared with DMSO-treated plus SVCV-infected cells.

**Figure 6 microorganisms-11-02812-f006:**
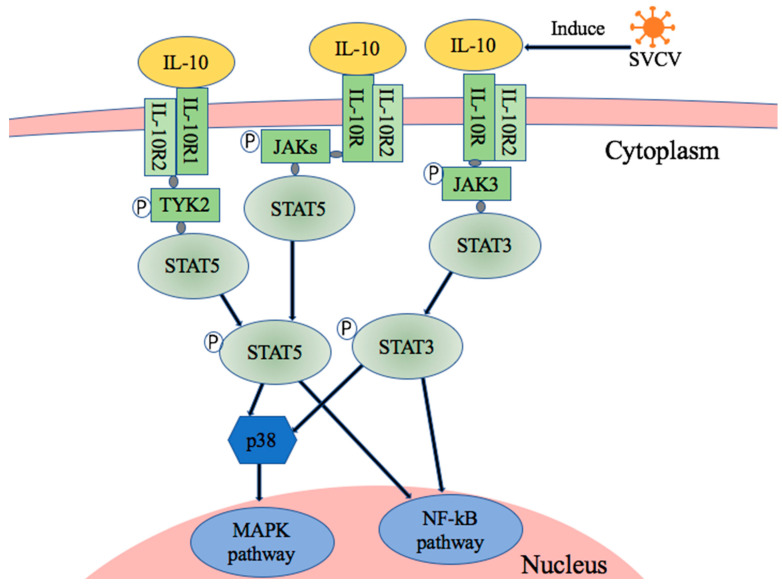
The potential signal pathway of SVCV-infection-induced carp IL-10 production.

**Table 1 microorganisms-11-02812-t001:** Primers used for RT-PCR [23,24].

Primers	Sequence (5′–3′)	Accession No.
β-actin-F	GACCTGTATGCCAACACTGTAT	M24113
β-actin-R	TCCTGCTTGCTAATCCACATC	
IL-10-F	AGTCCTTATGGCTGTCACGTCATG	AB110780
IL-10-R	TTGAGTGCAAGTGGTCCTTCTGG	
JAK1-F	AGGGGACACCTCTACTGGATGC	AH004872
JAK1-R	GTGTGAGAAGTTACGCTGCTTA	
JAK2-F	CGGAGTGTCACCAGTCTAC	KJ782027
JAK2-R	GGTAAGACATAACACAGTCATCC	
JAK3-F	GGCATCAGAGGACCTTTCATAC	AF148993
JAK3-R	CTGCCATTCCCAAGCATTCCTG	
TYK2-F	TGAGGGTGAGGTGACTGCTGAAG	KJ782031
TYK2-R	AGCGGTTCCTTTTCACCTAATCC	
STAT5-F	CCTGGGATGGAATAGATTTAGAG	KJ782032
STAT5-R	GACAGCGGTCATACGTGCTCTTTAG	
STAT6-F	AATGACCCCGCAGTTACAGTTC	KJ782033
STAT6-R	GTTGTAGTTTGACCCTCTCCAC	

## Data Availability

Data are contained within the article.

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
