# Peer review of "Spring Viremia of Carp Virus Infection Induces Carp IL-10 Expression, Both In Vitro and In Vivo"

_microorganisms, 2023, doi:10.3390/microorganisms11112812_

Round 1
Reviewer 1 Report
Comments and Suggestions for Authors
Before review, the manuscript needs major textual and language editing. I will be happy to complete a scientific review if the authors can improve the the text/language, but the current level is to far away from the standards expected in a scientific paper.
Comments on the Quality of English Language
Some comments from the introduction:
Line 18: insert “are” still unknown….
Line 19: change was with is
Line 32: what is a «mammal virus»?
Line 40: change was with is
Line 45: «killed». Consider changing to euthanized
Line 46: change founded with found. And what is “Th2 fine cells”?
Line 49: change was with is
Line 50: change were with are
Line 54: “so as to…” makes no sense
Line 56: change were with are
Line 59: “to enhancement” makes no sense
Line 60: “and so on” - redundant
Line 60: “there are also some viruses encode” makes no sense
Line 65: “had been studied” makes no sense
Line 67: “at present, many fish of” makes no sense. “Successively” makes no sense
Line 81: “mechanisms” and again “mammal viruses?”
This goes on for the entire manuscript. Please have a native speaker to proof read the manuscript
Author Response
Answer: Thank you very much for your suggestions. The manuscript has been reviewed by a native English speaker.
Question #1 Line 18: insert “are” still unknown
Answer: Agreed. It has been modified in the revised version at line18
Question #2 Line 19: change was with is
Answer: Agreed. It has been modified in the revised version at line19
Question #3 Line 32: what is a «mammal virus»?
Answer: Thank you for your comments. It has been replaced by mammalian viruses at line 33
Question #4 Line 40: change was with is
Answer: Agreed. It has been modified in the revised version at line41
Question #5 Line 45: «killed». Consider changing to euthanized
Answer: Agreed. It has been modified in the revised version at line45
Question #6 Line 46: change founded with found. And what is “Th2 fine cells”?
Answer: Thank you for your comments. It has been modified in the revised version at line47.
Question #7 Line 49: change was with is
Answer: Agreed. It has been modified in the revised version at line50
Question #8 Line 50: change were with are
Answer: Agreed. It has been modified in the revised version at line51
Question #9 Line 54: “so as to…” makes no sense
Answer: Agreed. It has been modified in the revised version at line54-55
Question #10 Line 56: change were with are
Answer: Agreed. It has been modified in the revised version at line57
Question #11 Line 59: “to enhancement” makes no sense
Answer: Agreed. It has been modified in the revised version at line59
Question #12 Line 60: “and so on” - redundant
Answer: Agreed. It has been modified in the revised version at line60
Question #13 Line 60: “there are also some viruses encode” makes no sense
Answer: Agreed. It has been modified in the revised version at line60
Question #14 Line 65: “had been studied” makes no sense
Answer: Agreed. It has been modified in the revised version at line63
Question #15 Line 67: "At the moment, many fish" makes no sense. "Sequentially" is meaningless
Answer: Agree. Modified in revision at line 66
Question #16 Line 81: “Mechanism” and “Mammalian viruses?
Answer: Thank you for your comment. Modified in revision line 81

Reviewer 2 Report
Comments and Suggestions for Authors
The authors of the manuscript microorganisms-2631573 describe the induction and expression in vivo and in vitro of interleukin 10 in infection by Spring Viraemia of Carp.
The paper is interesting and well detailed. There are some small typos to correct (line 69, 70 and 86 the scientific names in italics) and the description of the experimental design should be implemented to make it more easily interpretable. Furthermore, the chemical-physical conditions of the water during the experiment should be better reported and the number of subjects treated per tank and any replications should be promptly indicated.
The discussion correctly supports the results described and the bibliography is correctly inserted.
For these reasons, this paper must undergo only minor revisions.
Author Response
Question #1 There are some small typos to correct (line 69, 70 and 86 the scientific names in italics) The paper is interesting and well detailed. There are some small typos to correct (line 69, 70 and 86 the scientific names in italics)
Answer: Thank you for your comments. The manuscript has been reviewed by a native English speaker to improve the English.
Question #2 The description of the experimental design should be implemented to make it more easily interpretable
Answer: Thank you for your comments. It has been revised in the manuscript.
Question #3 Furthermore, the chemical-physical conditions of the water during the experiment should be better reported and the number of subjects treated per tank and any replications should be promptly indicated.
Answer: Thank you for your comments. The related content has been added in the revised manuscript and highlight in yellow.

Round 2
Reviewer 1 Report
Comments and Suggestions for Authors
«Spring Viraemia of Carp Virus Infection Induces Carp IL-10 2 Expression Both in vitro and in vivo» by Ouyang et al.
The study investigates mRNA and protein expression of IL10 in SVCV challanged and non-challanged carp cells and tissue. Methods include Rt-qPCR and ELISA assays. Inhibitors of IL10 upregulation were applied to investigate their respective function, providing in-depth information about the regulatory pathway. They conclude that the strategy of upregulation of IL10 seems equally important in viral infections in fish as in mammals.
A general comment, which was also underlined in my intial review, concerns the language. Although the manuscript has been improved, there are still many errors, both typografical and orthografical. I will not comment on the lanugage this time around, but please, the manuscript needs a thorough revision.
Heading: The heading is informative and descriptive.
Abstracts: main points are covered.
Introduction: Reads well, but the authors should add more relevant refereances on IL10 expression and viral infections in teleost fish. Several studies indicate a pivotal role of IL10 in viral infections: Collet et al. 2015 investigated the expression of IL10 in Atlantic salmon peripheral blood following experimental infection with infectious salmon anemia virus showing induction at a late stage of the infection. Also, Ingerslev et al. 2009 found a significant upregulation 24 days post intraperitoneal challenge with infectious pancreatic necrosis virus. Bjørgen et al. 2020 showed a significant downregulation of IL10 in the acute hemorrhagic muscle changes associated with Piscine orthoreovirus 1 infection.
Materials and methods:
- Line 86: how many fish were svcv isolated from? Which tissue? How? This needs further description.
- Line89: how many fish?
- Line 91: although described before, a brief description should be added here as well.
- Line 100: how many fish?
Results: The results are nicely presentet and the figures are appropriate. This is the strong point of this manusript, in my opinion. Well done.
Discussion:
The disussion needs a thorough revision. The first paragraph of the discussion is malplaced (line 292-304). Also the second paragraph seems unnessesary and not related to the results (line 305-312). I would delete these two paragraphs. The authors should discuss their own results according to how they were presented under «results».
It would also be interesting if the authors could discuss their results on the timeframe and which organs showing increased Il10 expression up agains the pathogenesis of Spring Viraemia. Does this make sence with the typical progression of disease, both on an organ (gills etc) and a cellular (macrophages etc.) level?
Comments on the Quality of English Language
Needs a thorough review of the language - not at an expected level for a scientific paper
Author Response
The point-by-point response to the reviewers' comments
Dear editors,
Thank you very much for concerning our manuscript entitled “Spring Viraemia of Carp Virus Infection Induces Carp IL-10 Expression Both in vitro and in vivo” (Manuscript ID: microorganisms-2631573). The reviewer comments are all valuable and very helpful for revising and improving our paper, as well as the important guiding significance to our manuscript. On behalf of my co-authors, we would like to express our great appreciation to the editor and reviewers, and hope that the correction will meet with approval.
Thank you and best regards. The main corrections in the revised manuscript have been highlighted in yellow and the responses to the reviewer’s comments are as following:
REVIEWER COMMENTS:
Reviewer #1
Comment 1: The study investigates mRNA and protein expression of IL10 in SVCV challanged and non-challanged carp cells and tissue. Methods include Rt-qPCR and ELISA assays. Inhibitors of IL10 upregulation were applied to investigate their respective function, providing in-depth information about the regulatory pathway. They conclude that the strategy of upregulation of IL10 seems equally important in viral infections in fish as in mammals. A general comment, which was also underlined in my intial review, concerns the language. Although the manuscript has been improved, there are still many errors, both typografical and orthografical. I will not comment on the lanugage this time around, but please, the manuscript needs a thorough revision.
Response: Thank you very much for constructive suggestions. The new version of the manuscript has been revised by a native English-speaker.
Comment 2: Heading: The heading is informative and descriptive.
Response: Thank you very much for your comments.
Comment 3: Abstracts: main points are covered.
Response: Thank you very much for your comments.
Comment 4: Introduction: Reads well, but the authors should add more relevant refereances on IL10 expression and viral infections in teleost fish. Several studies indicate a pivotal role of IL10 in viral infections: Collet et al. 2015 investigated the expression of IL10 in Atlantic salmon peripheral blood following experimental infection with infectious salmon anemia virus showing induction at a late stage of the infection. Also, Ingerslev et al. 2009 found a significant upregulation 24 days post intraperitoneal challenge with infectious pancreatic necrosis virus. Bjørgen et al. 2020 showed a significant downregulation of IL10 in the acute hemorrhagic muscle changes associated with Piscine orthoreovirus 1 infection.
Response: Thank you very much for providing relevant references. These references have been cited in the revised version.
Comment 5: Materials and methods:
-Line 86: how many fish were svcv isolated from? Which tissue? How? This needs further description.
Answer: Thank you for your comments. The SVCV strain was isolated from the kidney and spleen mixture of a disease Percocypris pingi by using EPC cells in Chengdu, China, in 2018. It has been clarified in the revised manuscript.
-Line89: how many fish?Answer: Thank you for your comments. Four healthy common carp were used for isolate carp head kidney primary cells (cHK). It has been modified in the revised version at line93
-Line 91: although described before, a brief description should be added here as well.
Answer: Thank you for your comments. A brief description has been added in the revised version at Lines 95-102.
-Line 100: how many fish?
Answer: Thank you for your comments. Sixty common carp were used. It has been modified in the revised version at line107
Comment 6: Results: The results are nicely presentet and the figures are appropriate. This is the strong point of this manusript, in my opinion. Well done. Response: Thank you very much for your comments.
Comment 7: Discussion: The disussion needs a thorough revision. The first paragraph of the discussion is malplaced (line 292-304). Also the second paragraph seems unnessesary and not related to the results (line 305-312). I would delete these two paragraphs. The authors should discuss their own results according to how they were presented under «results». It would also be interesting if the authors could discuss their results on the timeframe and which organs showing increased Il10 expression up agains the pathogenesis of Spring Viraemia. Does this make sence with the typical progression of disease, both on an organ (gills etc) and a cellular (macrophages etc.) level?
Response: Thank you very much for constructive suggestions. We are agreed that it is interesting to discuss the difference expression of IL-10 on the timeframe and the organs. The discussion has been added in the discussion part.

Round 3
Reviewer 1 Report
Comments and Suggestions for Authors
The improvements are sufficient